# A High Precision Time Grating Displacement Sensor Based on Temporal and Spatial Modulation of Light-Field

**DOI:** 10.3390/s20030921

**Published:** 2020-02-09

**Authors:** Min Fu, Changli Li, Ge Zhu, Hailin Shi, Fan Chen

**Affiliations:** 1Engineering Research Center of Mechanical Testing Technology and Equipment, Ministry of Education, Chongqing Key Laboratory of Time Grating Sensing and Advanced Testing Technology, Chongqing University of Technology, Chongqing 400054, China; li_985483353@163.com (C.L.); shl0605hln@163.com (H.S.); chen18827512638@163.com (F.C.); 2The College of Mechanical Engineering, Chongqing University of Technology, Chongqing 400054, China; gezhuge@cqut.edu.cn

**Keywords:** optical measurement, displacement sensor, light field distribution, scattering error

## Abstract

A new displacement sensor with light-field modulation, named as time grating, was proposed in this study. The purpose of this study was to reduce the reliance on high-precision measurements on high-precision manufacturing. The proposed sensor uses a light source to produce an alternative light-field simultaneously for four groups of sinusoidal light transmission surfaces. Using the four orthogonally alternative light-fields as the carrier to synthesize a traveling wave signal which makes the object movement in the spatial proportion to the signal phase shift in the time, the moving displacement of the object can be measured by counting time pulses. The influence of the light-field distribution on sensor measurement error was analyzed in detail. Aimed to reduce these influences, an optimization method that used continuous cosinusoidal light transmission surfaces with spatially symmetrical distribution was proposed, and the effectiveness of this method was verified with simulations and experiments. Experimental results demonstrated that the measurement accuracy reached 0.64 μm, within the range of 500 mm, with 0.6 mm pitch. Therefore, the light-field time grating can achieve high precision measurement with a low cost and submillimeter period sensing unit.

## 1. Introduction

With the development of precision and ultra-precision manufacturing technology, the accurate µm and nm displacement measurements are receiving greater attention. Typical methods for measuring large displacement are using a laser interferometer or a grating [1,2,3]. The former, which uses laser wavelength as a unit of measurement, has high measurement accuracy, but is greatly disturbed by the external environment, and is suitable for use in well-controlled environments [4,5]. The latter, which uses a structure of periodic hyperfine grating line as the measurement standard, has more robust to measurement environment and is primarily used in the workshop applications [6,7]. A measurement with a grating uses Moiré fringes formed by superimposing periodic grid lines between a scale grating and index grating. Because its measurement datum is based on the period of the grating scale [8], a grating with smaller pitch provides higher measurement accuracy. Nowadays, many researchers have proposed many lithography methods to reduce the grating period, such as multiple nanoimprint lithography [9], pressed self-perfection by liquefaction [10], and negative electron-beam lithography [11]. However, with the further reduction of the grating pitch, the lithography manufacturing process will approach the diffraction limit [12,13], beyond which the grating pitch cannot be reduced further. In order to reduce the difficulty in grating manufacturing and to increase the measurement accuracy, many scholars have investigated grating subdivision technology, such as electronic subdivision [14,15], phase subdivision [16] and optical subdivision [17]. Because the grating pitch is very small, the quality of the electrical signal is affected by the manufacturing process and collimation of the light source. Some researchers had analyzed the effect of scattering on Moiré fringe formation [18,19,20] and proposed some optimized schemes [21,22]. However, because the effect of the light source on measurement is not the main reason and the collimation of the light source is very limited [23,24], the measurement accuracy of the grating still relies on the precision of the grating manufacturing process. This relationship between measurement accuracy and manufacturing process limits the further improvement of grating measurement accuracy.

In order to eliminate the extreme dependence of highly accurate measurements of high precision manufacture, Liu [25,26,27,28] developed an electric field time grating with a submillimeter pitch to provide large displacement measurements with nm accuracy. The electric field time grating used an alternating electric field and a capacitor array structure to form a motion coordinate system, which reduced manufacturing difficulty and achieved high-precision measurement. However, the installation distance between the fixed ruler and the moving ruler would directly affect the measurement accuracy of the electric field time grating, thereby affecting its further applications in the engineering field.

To further improve the time grating measurement method, we propose an optical time grating sensor that uses temporal and spatial intensity modulation to achieve accurate displacement measurements. Because the principle of this sensor is based on the precise modulation of light intensity, which is similar to magnetic or electric field modulation, we call this sensor a light field time grating. The influence of the distance between two rulers on the measurement accuracy in the electric field time grating can be eliminated by using light-field as the measurement medium. The measurement principle of the proposed sensor is based on modulation of the light-field. Thus, the measurement accuracy is no longer determined by the high precision manufacturing. In this paper, the characteristics of the light-field distribution of different scattering angles are analyzed, and the mathematic error model and the simulation model of the scattering angle are established. An optimization method of continuous cosinusoidal light transmission surfaces with spatially symmetrical distribution is proposed to obtain a high-precision modulation of the light-field. Experimental results show that the optimization method can effectively reduce the main period error caused by light field distribution to achieve high accuracy measurement.

## 2. Structure and Measurement Principle

The structure of the light field time grating is illustrated in Figure 1a, which consists of a light source, moving ruler, fixed ruler, and photodetector. As shown in Figure 1a, the light source, moving ruler, and the photodetector are fixed together. The moving ruler consists of a row of equally spaced rectangular light transmission surfaces, and the fixed ruler consists of four groups of sinusoidal transmissive surfaces whose spatial phase are offset from each other by *π*/2 (equal to the distance of *W*/2). For convenience, the four groups of light transmission surfaces are named 0°, 90°, 180°, and 270° surface, respectively. The width of the rectangular surfaces of the moving ruler is equal to that of the sinusoidal surfaces of the fixed ruler. The photodetector consists of four groups of photocells that receive the four groups of light transmitting signals, respectively. Just like an alternating magnetic field generated by an alternating current signal, an alternating light-field signal for the four groups of light transmission surfaces is provided by the light source, which is excited by an alternating current signal. When the moving ruler moves relative to the fixed ruler, the light transmission area of the moving ruler will change sinusoidally, as shown in Figure 1b. This spatially modulated light transmission area change is directly related to the measured object displacement. Taking the first group of light transmission surfaces (0° surface) in Figure 1b as an example, when the moving ruler moves by *x* displacement from the fixed ruler as shown in Figure 1c, the light transmission area is a section surrounded by the sinusoidal surface profile and the displacement *x*.

The relationship between the sinusoidal surface profile (red light transmission surface profile in Figure 1) and the displacement *x* is described as follows:(1)y=Asin(πx/W)  x⊂[0,2W]
where *A* is the height of the sinusoidal light transmission area; *W* is the width of the sinusoidal light transmission surface. The sinusoidal light transmission area is the integral of the Equation (1) and the displacement *x*. The relationship between the light transmission area and the displacement is expressed as follows:(2)$$S{\left(x\right)}_{0{}^{\circ}}=2\int_{0}^{x}A\sin(\pi x/W)dx=(2WA/\pi )\left[1-\cos(\pi x/W)\right]\;\;x\subset [0,2W]$$

Equation (2) shows that the light transmission area exhibits a cosine law variation with the displacement *x*. In the same way, the other three groups of light transmission surfaces also have the same variation, but the spatial phase is different from each other by π/2, which are:(3)$$\left\{ \begin{array}{l} S{\left(x\right)}_{90{}^{\circ}}=(2WA/\pi )[1-\cos(\pi x/W-\pi /2)]=(2WA/\pi )[1-\sin(\pi x/W)]\\ S{\left(x\right)}_{180{}^{\circ}}=(2WA/\pi )[1-\sin((\pi x/W-\pi /2)]=(2WA/\pi )[1+\cos(\pi x/W)]\\ S{\left(x\right)}_{270{}^{\circ}}=(2WA/\pi )[1+\cos((\pi x/W-\pi /2)]=(2WA/\pi )[1+\sin(\pi x/W)] \end{array}\right.\;\;x\subset [0,2W]$$

Combined with Equations (2) and (3), the light transmission areas of four groups of light transmission surfaces exhibit sinusoidal variation which can be defined as four spatial frequency signals with π/2 spatially phase offset of each other. Four groups of photocells, respectively, receive the optical signals of the four groups of light transmission surfaces. This means that the optical signals received by the four groups of photocells will also exhibit the same relationship change in the spatial domain. If the light source is excited by an alternating current signal with frequency *ω*, the four groups of photocells will obtain alternating light intensity signals in the time domain. The functions of these four temporal and spatial modulated light signals can be expressed as follow:(4)$$I_{i}=I_{m}(1+\cos\omega t)S{\left(x\right)}_{i}\;\;(i=0{}^{\circ},\;90{}^{\circ},\;180{}^{\circ},\;270{}^{\circ})$$
where *i* is one of the four signal channels, whose relative phases are equal to 0°, 90°, 180°, and 270°; *I_m_* refers to the maximum amplitude of the alternating light intensity; *I_i_* refers to the *i-*th channel’s signal. Combined with Equations (2)–(4), the signal of Ii is composed of a standing wave signal and a spatial frequency signal.

Figure 2 contains a block diagram showing light-field time grating measurements. Four photocells are used to receive the four light signals (*I*_0°_, *I*_90°_, *I*_180°,_ and *I*_270°_), as shown in Figure 2a. After conversion by photoelectric acquisition circuit, the four corresponding voltage signals are *U*_0°_, *U*_90°_, *U*_180°_, and *U*_270°_. A standing wave signal (named *U*_13_) is achieved by subtracting *U*_0°_ from *U*_180°_ to eliminate the direct-current (DC) component. Similarly, another standing wave signal (named *U*_24_) is obtained by subtracting *U*_90°_ from *U*_270°_. The *U*_13_ and *U*_24_ have the same temporal phase and the orthogonal spatial phase. By taking one of the two signals (such as *U*_13_) for the π/2 phase shift, two temporal and spatial orthogonal standing wave signals can be obtained, named *U*_13′_ and *U*_24_. A traveling wave signal *U* can be obtained by superimposing the two signals and then passing the high-pass filter to filter out the low-frequency components, as shown in Figure 2b.

The sum of *U*_13′_ and *U*_24_ is:(5)$$U_{0}=U_{13^{\prime }}+U_{24}=(4AWI_{m}/\pi )\{[\cos(\pi x/W)+\sin(\pi x/W)]+\sin\left(\omega t+\pi x/W\right)\}$$

Equation (5) consists of two spacial frequency signals (the value is periodically varied with the spatial position) and a traveling wave signal. Using a high-pass filter eliminates the spatial frequency signals, yielding the following traveling wave signal *U*:(6)$$U=(4AWI_{m}/\pi )\sin\left(\omega t+\pi x/W\right)$$

Equation (6) describes a traveling wave signal with frequency *ω*. The phase value of *U* reflects the displacement of the measured object. As shown in Figure 2), using the exciting signal as the reference signal *U_r_*, the linear displacement of the moving ruler is described by the phase difference between *U* and *U_r_*, which can be measured by detecting the time difference of the zero-crossing points. As the time difference can be determined by counting high-frequency time pulses, the displacement is measured by counting time pulses, and the sequence of time pulses is considered as a grating of time for displacement measurement. Therefore, this displacement sensing method is called time grating.

## 3. Light-Field Distribution Characteristics and Error Analysis

### 3.1. The Light-Field Distribution Characteristics Analysis

The most prominent feature of the light-field time grating is using the temporal and spatial modulated light-field to achieve displacement measurement. Because the electric system can well promise the accuracy of temporal modulation, the light-field distribution of the spatial modulation is the main factor affecting the accuracy of the measurement. Usually, the uniformity of the light intensity and the divergence angle of the light source are two main aspects of the light-field distribution.

Figure 3 compared several light sources with different divergence angles and light intensity distributions. Theoretically, when the divergence angle is 0°, and intensity distribution is very uniformity, there is no measurement error caused by light-field distribution, as shown in Figure 3a. However, an ideal parallel light source does not exist, and the actual light-field distributions are shown in Figure 3b,c.

Firstly, it can be seen from the light intensity distribution curve provided by the light source manufacture that the light intensity distribution in the central region is very uniform and in the surrounding area presents a concentric distribution, and as the scattering angle increases, the uniformity of the light intensity distribution deteriorates. If the transmission surfaces are arranged in the central area of the light spot, the characteristic of uniformity of the light intensity can be made fully utilize, and the influence of uniformity of light intensity distribution may be effectively solved. Secondly, the divergence angle will change the shape of the light transmission areas, which becomes more blurred as the light divergence angle increases, as shown in Figure 3b,c. When the light source’s half-angle equal to 5.5°, the shape of the light transmission areas become very blurred and difficult to distinguish. Therefore, the measurement errors of the sensor caused by light-field distribution mainly come from light scattering. We will focus on analyzing the effect of the divergence angle on measurement accuracy in the next paper.

### 3.2. Scattering Model and Error Analysis

A collimated LED is equivalent to a point source with a very long focal length [29,30,31]. Figure 4a contains a schematic diagram showing a scattering model of light from a collimated LED for a certain divergence angle. The width of the light transmission surfaces is changed by scattering and refraction of the emitted light source.

Figure 4a shows that the width of the light transmission area varies with the incident angle and the variations in the width of the area is bigger when the incident angle is larger. The central symmetry feature of the point source makes the light transmission areas on both sides of the centerline exhibit opposite variations. This phenomenon will cause an offset in the spatial phase shift. The area to the left of the centerline is phase-advanced, and the area to the right of the centerline is phase-lagged. It means that under the influence of the incident angle, the spatial phase of the light transmission areas on the left side of the centerline is ahead of the ideal phase, and the light transmission areas on the right side of the centerline are lagging the ideal condition.

If the four groups of light transmission surfaces are assigned on two sides of the centerline as shown in Figure 1b, the value of the phase shift of the light transmission surfaces located in the two sides of centerline will have opposite changes. In fact, due to the different incident angles of light at different positions, the phase shift from each light transmission surfaces will be different, but the light signal from the same group of light transmission surfaces is the sum of all light transmission surfaces’ areas. Therefore, for convenience, we can use an average phase shift to define the value of phase offset from the same group of light transmission surfaces. Assuming that the light transmission surfaces on the right of centerline have a positive average phase offset Δ*x*_0_, as shown in Figure 4b, the light transmission area can be written as follows:(7)$$S\left(x\right)=2\int_{0}^{x+\Delta x}A\sin(\pi x/W)dx=(2WA/\pi )\{1-\cos[\pi (x+\Delta x)/W]\}$$
where *x* is the actual displacement of the moving ruler. If the light transmission surfaces are arranged as shown in Figure 1b, the light transmission surfaces of 0° and 180° assigned on the right of centerline have a positive phase shift of Δ*x*, and the light transmission surfaces of 90° and 270° assigned on the left of centerline have a negative phase shift of −Δ*x*. Similar to the analytical method of Equations (2) and (3), the actual area variation for each group of light transmission surfaces can be written as follows:(8)$$\left\{ \begin{array}{l} S^{\prime }{\left(x\right)}_{0{}^{\circ}}=2\int_{0}^{x+\Delta x}A\sin(\pi x/W)dx=(2WA/\pi )\{1-\cos[\pi \left(x+\Delta x\right)/W]\}\\ S^{\prime }{\left(x\right)}_{90{}^{\circ}}=(2WA/\pi )\{1-\sin[\pi \left(x-\Delta x\right)/W]\}\\ S^{\prime }{\left(x\right)}_{180{}^{\circ}}=(2WA/\pi )\{1+\cos[\pi \left(x+\Delta x\right)/W]\}\\ S^{\prime }{\left(x\right)}_{270{}^{\circ}}=(2WA/\pi )\{1+\sin[\pi \left(x-\Delta x\right)/W]\} \end{array}\right.$$

It can be seen that the actual light transmission area will have a spatial phase change, due to the influence of light source scattering. The traveling wave signal affected by the scattering of the light source is expressed as follows:(9)$$\begin{array}{ll} U & =K_{1}\sin(\omega t)\cos[\pi (x+\Delta x)/W]-K_{1}\cos(\omega t)\sin[\pi (x-\Delta x)/W]\\ & =K_{1}\sqrt{{\left[\cos\frac{\pi }{W}(x+\Delta x)\right]}^{2}+{\left[\sin\frac{\pi }{W}(x-\Delta x)\right]}^{2}}\sin\left(\omega t-\arg tg\frac{\sin\frac{\pi }{W}(x-\Delta x)}{\cos\frac{\pi }{W}(x+\Delta x)}\right) \end{array}$$
where $K_{1}=4AWI_{m}/\pi$. The Equation (9) shows that both the amplitude and the phase of the traveling wave signal are changed, but it is known from the measurement principle of the proposed sensor that the measurement error is determined by the phase error in the traveling wave signal, so the measurement error in Equation (9) can be expressed:(10)e(x)=arctan{sin[π(x−Δx)/W]/cos[π(x+Δx)/W]}−arctan[tan(πx/W)]

Equation (10) is a complex inverse trigonometric function and hard to directly analyze the error relationship, but the effect of phase shift Δ*x* can be simulated by software. Assuming the width of the light transmission surface is 0.6 mm, the error curve in Equation (10) can be simulated by using Matlab as shown in Figure 5a. We can see that the variation in error is periodic and mainly represents a second-order error in the range of one period (0.6 mm). Moreover, the measurement error increases significantly as the phase shift Δ*x* increases. Using Fourier spectrum analysis on the error curve, as shown in Figure 5b, the result shows that the primary error components caused by scattering are the second-order and fourth-order harmonic components, of which the second-order error is dominant.

## 4. Experiments and Discussion

The experimental system was shown in Figure 6. The moving ruler with a length of 530 mm and 0.6 mm period was fastened on the moving part of the AEROTECH precision linear platform. The moving ruler could be driven with a linear motor on the precision linear platform. This linear platform has a positioning accuracy of 1.1 μm in the range of 1200 mm and a minimum control step of 0.1 μm, which can satisfy the requirements of the proposed sensor performance testing. The LED, photodetector, and fixed ruler were fastened on a readhead, which was mounted on a multi-degree-of-freedom fine-tuning mechanism to adjust the relative position between the fixed ruler and the moving ruler. The laser interferometer Renishaw XL-80 with a ± 0.5 ppm precision was used as a standard reference system to calibrate the light field time grating measurements.

From the previous analyses, we know that the light source scattering and light intensity distribution are two main influence factors on the measurement error. However, the uniform distribution of the light intensity distribution can be selected on the light transmission surface arrangement to avoid the influence, so the influence of the light source scattering on the measurement will be mainly verified next.

### 4.1. Experiment of the Divergence Error

In order to examine the influence of scattering on measurement errors, two light sources with half divergence angles of 1.8° and 5.5° were used for comparative experiments, as shown in Figure 3. The other parameters in the two experiments remained the same, including the same sensor prototype, mounting structure, photodetector, and signal processing system. During the experiment, the linear motor was used to drive the moving ruler in 0.01 mm steps, and measurements were gathered at 60 different positions within one period (0.6 mm). The measurement error profile can be obtained by subtracting the value measured by the proposed sensor from that measured with the laser interferometer. The error curves and harmonic components in the two experiments are shown in Figure 7.

Figure 7a,b shows that the measurement errors are primarily from the first to the fourth harmonic errors, of which the second-order is dominant. Furthermore, as the angle of light scattering increases, the measurement error is increased from 0.79 μm to 1.07 μm within one period, and the second-order and the fourth-order harmonic errors increase by 0.14 μm and 0.1 μm. Therefore, the influence of the scattering angle on measurement error is primarily reflected in the second-order and the fourth-order harmonic errors. This is consistent with the previous theoretical analysis and simulation results, as shown in Figure 5. In addition, as analyzed in the previous paper, because the measurement principle of the traditional grating sensor is based on the intensity change of the rectangle grating, the error components of the measurement are very complicated and very difficult to eliminate from principle [32,33]. Compared with the traditional grating, those low frequency and periodic errors of the proposed measuring method in the Figure 7 are easier to correction from the source of errors.

### 4.2. Discussion

It is easy to find from Figure 7 that the second-order and fourth-order harmonic errors were still the main error components, so the scattering was still the most influential factor in the measurement results. Combined with the previous analysis, the main effect of scattering on measurement can be divided into twofold. On the one hand, as the divergence angle increases, the light transmission surfaces become more blurred meaning the effective luminous flux decreases. On the other hand, the divergence angle of the light source produces different phase shifts at the light transmission areas; thus, it has a tremendous influence on the measurement.

To deeply analyze the influence of scattering, a simulation model for the proposed sensor was constructed using TrancePro, and the model was shown in Figure 8a. Using the 5.5° light intensity distribution curve in Figure 3, the parameters of the light source are set in the software, so that the light intensity distribution and the scattering angle of the light source for the simulation are similar to the real situation. The simulation effect of the light source is shown in Figure 8b. The other parameters in Figure 8a are L=8 mm, γ=0 mm, δ=0.5 mm, D=0.75 mm, W=0.3 mm. Finally, the contrast simulation of the luminous flux in one period of different kinds of light transmission surfaces can be carried out.

Using the sinusoidal transmission surfaces of Figure 1 as a research object, the simulation of the light intensity distribution of the sinusoidal light transmission areas is shown in Figure 9a. The contour of the sinusoidal surfaces becomes blurred, which will cause the width of light transmission surfaces greater than the width of the opaque surfaces. This phenomenon will prevent the varying of the spatial modulated area from changing with a sinusoidal law so that the received spatial frequency signals will also not change in the law of sine. For the further understanding of how much influences of the scattering on the spatial modulation would be caused, the luminous flux change relationship of the four groups of light transmission surfaces is simulated within one spatial period (0.6 mm), as shown in Figure 9b. From the simulation results, we can see that the four spatial modulated signals are not changed with the rule of sinusoidal. Since the luminous flux signal can directly reflect the effects of spatial modulation of transmission surfaces, the four spatial frequency signals received by photodetector are also not changed with the rule of sinusoidal. Further, by multiplying these four signals by the time modulation signal, respectively, a traveling wave signal based on the sensing mechanism can be obtained, as shown in Figure 9c. The amplitude of the traveling wave is not a constant that varies with displacement. Using Fourier spectrum analysis on the amplitude envelope curve, the result shows that the mainly periodic errors contain the second and fourth harmonic error components, which agree with the previous analysis of Figure 5.

According to the above analysis, the sinusoidal light-transmitting surface with an equidistant distribution will directly affect the quality of the spatially modulated signal. Thus, instead of the disperse sinusoidal surfaces, we proposed a continuous cosin transmission surfaces. The structure and simulated affection are shown in Figure 9d. This new structure can bring two benefits to overcome the influence of the scattering. Firstly, although the scattering also makes the transmission area blur, this continuous structure can promise the change of transmission area along with the sinusoidal rule. Since the contour of cosine transmission areas continuously changes within one period, the value of the light transmission areas can continuously change from the maximum value to the minimum value with a sine law during this process. Secondly, compared to the disperse sinusoidal surfaces, the luminous flux of the continuous cosine surfaces is increased, thereby improving the Signal-to-Noise-Ratio (SNR) of the spatially modulated signal. Similar to the above analysis, the luminous flux of four groups of transmission surfaces and the traveling wave signal is simulated, as shown in Figure 9e,f. The simulation results show that both the sine curve of the luminous flux and the amplitude of the traveling wave signal have been improved, but the fluctuation of the traveling wave still exists.

In order to further improve the effect of light scattering on the phase shift, a symmetrical continuous cosine structure is proposed based on the above analysis, as shown in Figure 10a. Because the phase shift of the light scattering has the characteristic of phase advance and phase lag on two sides of the light spot center, the symmetrical distribution light transmission surfaces can compensate for this phase offset. Taking the 0° light transmission surfaces as an example, the six surfaces are symmetrically distributed on two sides of the centerline, which have opposite phase shifts Δx and −Δx, as shown in Figure 10b.

Similar to the analysis method of Equation (7), the relationship in the light transmission area can be expressed as follows:(11)$$\begin{array}{ll} S^{\prime \prime }{\left(x\right)}_{0{}^{\circ}} & =S{\left(x\right)}^{+}+S{\left(x\right)}^{-}\\ & =\left[2\int_{0}^{x+\Delta x}A\left(1-\cos\frac{\pi }{W}x\right)dx+2\int_{0}^{W-x-\Delta x}A\left(1-\cos\frac{\pi }{W}x\right)dx\right]+\left[2\int_{0}^{x-\Delta x}A\left(1-\cos\frac{\pi }{W}x\right)dx+2\int_{0}^{W-x+\Delta x}A\left(1-\cos\frac{\pi }{W}x\right)dx\right]\\ & =4AW\left[1-2/\pi \sin\left(\pi x/W\right)\cos\left(\pi \Delta x/W\right)\right] \end{array}$$

Because the four groups of light transmission surfaces are offset from each other by a spatial phase of π/2, the area of the remain three groups of light transmission surfaces can express as follows:(12)$$\left\{ \begin{array}{l} S^{\prime \prime }{\left(x\right)}_{90{}^{\circ}}=4AW\left[1-\frac{2}{\pi }\sin\left(\frac{\pi }{W}x-\frac{\pi }{2}\right)\cos\left(\frac{\pi }{W}\Delta x\right)\right]=4AW\left[1+\frac{2}{\pi }\cos\left(\frac{\pi }{W}x\right)\cos\left(\frac{\pi }{W}\Delta x\right)\right]\\ S^{\prime \prime }{\left(x\right)}_{180{}^{\circ}}==4AW\left[1+\frac{2}{\pi }\cos\left(\frac{\pi }{W}x-\frac{\pi }{2}\right)\cos\left(\frac{\pi }{W}\Delta x\right)\right]=4AW\left[1+\frac{2}{\pi }\sin\left(\frac{\pi }{W}x\right)\cos\left(\frac{\pi }{W}\Delta x\right)\right]\\ S^{\prime \prime }{\left(x\right)}_{270{}^{\circ}}=4AW\left[1+\frac{2}{\pi }\sin\left(\frac{\pi }{W}x-\frac{\pi }{2}\right)\cos\left(\frac{\pi }{W}\Delta x\right)\right]=4AW\left[1-\frac{2}{\pi }\cos\left(\frac{\pi }{W}x\right)\cos\left(\frac{\pi }{W}\Delta x\right)\right] \end{array}\right.$$

Using the same analytical method of Equations (2)–(6), the signal synthesized by the two standing wave signals can be expressed as follows:(13)$$\begin{array}{l} U_{0}^{\prime }=U_{13}^{\prime }-U_{24}\\ =\frac{8AWI_{m}}{\pi }\left[\left(\sin\frac{\pi }{W}x\cos\frac{\pi }{W}\Delta x-\cos\frac{\pi }{W}x\cos\frac{\pi }{W}\Delta x\right)\right]-\frac{8AWI_{m}}{\pi }\cos\left(\frac{\pi }{W}\Delta x\right)\sin\left(\omega t+\frac{\pi }{W}x\right) \end{array}$$
where *U*^′^_13_ and *U*_24_ are two standing wave signals in Figure 2. The Equation (13) consists of two parts, two spatial frequency signals in front of the minus sign, and one amplitude-varying traveling wave behind the minus sign. After the spatial frequency signals are removed by a high-pass filter, the traveling wave signal can be obtained:(14)$$U^{\prime }=(-8AWI_{m}/\pi )\cos\left(\pi \Delta x/W\right)\sin\left(\omega t+\pi x/W\right)$$

The Equation (14) shows that the effect of the scattering on the phase shift is converted into the amplitude variation of the traveling wave. Since the displacement of the measured object is determined by the phase of the traveling wave signal, the phase shift caused by light scattering can be eliminated in Equation (14).

As shown in Figure 10c,d, using the same simulation method as Figure 9, the luminous flux of the four groups of transmission surfaces and the traveling wave signal of this symmetric structure were simulated. Compared with Figure 9e,f, the quality of the four spatial modulated signals is further improved, and the fluctuation of the traveling wave is also well suppressed. The simulation results demonstrate that this symmetrical structure may have the ability to overcome the influence of light scattering.

### 4.3. Optimized Design and Experimental Verification

Combined with the previous analysis, a new fixed ruler composed of continuous cosinusoidal light transmission surfaces and the symmetric structure was designed and fabricated to reduce the influence of the light-field distribution on the measurement accuracy, as shown in Figure 11a. As shown in Figure 10a, a total of eight groups of light-transmitting surfaces were arranged symmetrically. The other four surfaces were used as flags of installation between the fixed ruler and moving ruler to ensure that the two rulers were installed in parallel. An integrated photodetector consisting of 12 independent photocells was used to measure the light transmission signals, of which eight channels were used to receive the eight modulated light intensity signals, while the other four channels were used for receiving the installation signals, as shown in Figure 11b.

Using this optimized fixed ruler, a comparative experiment was performed using two previously used light sources on the same experimental platform shown in Figure 6. The measurement error curves were shown in Figure 12.

After comparing Figure 12 with Figure 7, one can see that the dominant factors determining measurement error are the first, second, and third-order harmonic errors. The total error value was significantly reduced, and fourth-order harmonic errors were basically eliminated. When the light scattering angle equals 1.8°, the measurement error was reduced from 0.79 to 0.4 μm within the 0.6 mm. When the scattering angle equals to 5.5°, the measurement error was reduced from 1.07 to 0.5 μm. In addition, Figure 12a,b shows that the measurement error had increased by 0.1 μm, and the second-order harmonic error had hardly changed. According to the previous analysis, the scattering angle mainly causes the second-order and fourth-order harmonic errors of the measurement error. The experimental results verify that the optimized fixed ruler can well suppress the error of light source scattering. At the same time, this symmetrical structure makes full use of the characteristics of the concentric circle distribution of light intensity, so that the influence caused by the uneven distribution of the light-field can also be better suppressed. The comparative experimental results show that the optimized design greatly improves the measurement accuracy, which is consistent with the theoretical and simulation analysis.

Finally, the linear motor was used to drive the moving ruler forward in 1 mm steps, and 500 position measurements were sampled (500 mm total distance). The measurement error curve is shown by the blue error curve in Figure 13, where the peak-to-peak value of the error is 2.4 μm, and the width of the error band is approximately 0.4 μm. The primary error components are caused by a low-frequency error over the full range, which is caused by the installation error and the manufacturing error. Since this long period error is a systematic error and has a fixed pattern of variation within the full range, this low-frequency error can be eliminated after calibrating the measured value with the laser interferometer measurement and repeatedly adjusting the sensor mounting structure. After error correction, the final measurement accuracy reaches 0.64 μm for a range of 500 mm, as shown by the red error curve in Figure 13.

## 5. Conclusions

In this paper, a low cost and highly accurate measurement method based on the modulation of light-field were proposed. In order to achieve high accuracy measurement with light-field modulation, the influence on the light-field distribution was divided into two parts: Light intensity uniformity and light scattering angle. The concentric circle characteristics of the light intensity distribution and the phase shift characteristics of the light source scattering were summarized, and the influence of these characteristics on measurement errors was analyzed. The correctness of the analysis of the influence of the light-field distribution on the measurement errors was proved by experiments. Further, through the contrast experiments, the effectiveness of the symmetrical cosine structure to suppress the influence of the light-field distribution was verified, and the accuracy of measurement was improved significantly. The results show that when the half divergence angle of the light source is 1.8°, the measurement accuracy within one period (0.6 mm) reached ± 0.2 μm and the measurement accuracy over the full range of 500 mm reached 0.64 μm after calibration. Compared with the traditional grating sensors with a pitch of micrometer-scale (< 20 μm), the present study uses a millimeter-scale grating surface modulation to achieve sub-micron measurement accuracy in a large measurement range. The manufacturing and cost of the sensor for high-precision measurements could be greatly reduced. Considering potential commercial applications, the effect of light-field distribution must be further reduced to improve the measurement accuracy. The resolution and measurement accuracy can be maintained at desired levels by adjusting the shape of the transmission surfaces in the future. Due to the novelty of the sensing method that adopts temporal and spatial modulation of light-field (compared to grating in the manufacturing), it has lower requirements on manufacturing cost and technology. Therefore, it has better application potential.

## Figures and Tables

**Figure 1 sensors-20-00921-f001:**
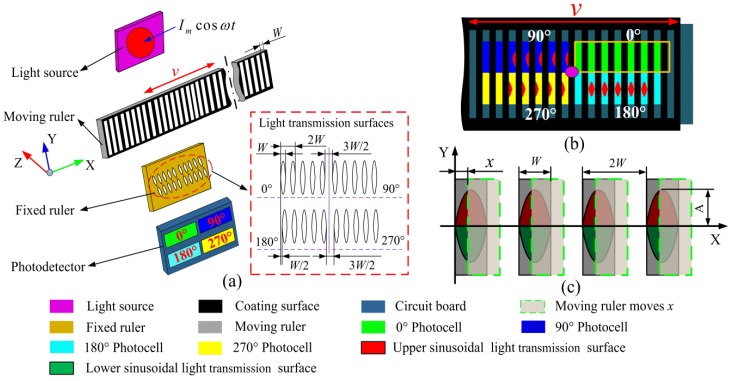
The structural model of the light-field time grating. (**a**) The structural diagram; (**b**) the positional relationship between the fixed ruler and the moving ruler; (**c**) the area change of sinusoidal light transmission surfaces.

**Figure 2 sensors-20-00921-f002:**
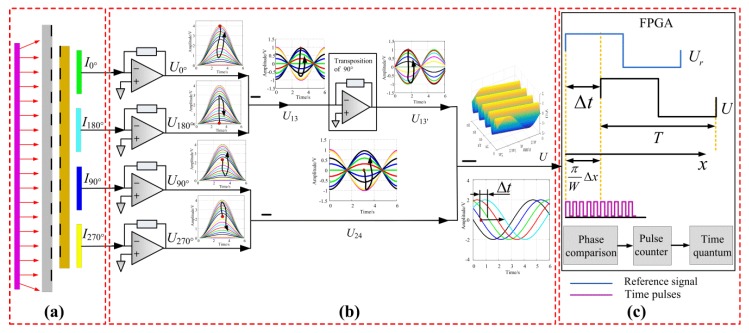
The measurement principle block diagram of the light-field time grating. (**a**) Photodetectric acquisition; (**b**) traveling wave synthesis; (**c**) relationship of displacement and time pluses.

**Figure 3 sensors-20-00921-f003:**
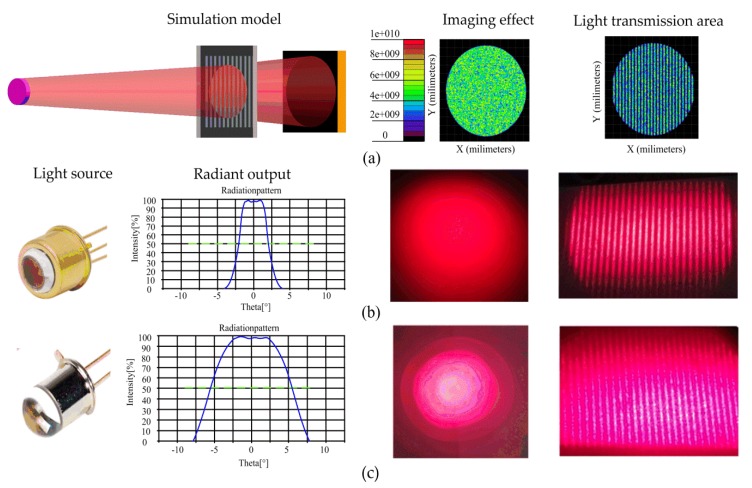
The characteristics of light sources. It includes a light intensity distribution curve, image of the light sources, and image of light transmission areas. (**a**) The light source of $\theta_{1/2}=0{}^{\circ}$ by simulation; (**b**) the light source of $\theta_{1/2}=1.8{}^{\circ}$ by measurement; (**c**) the light source of $\theta_{1/2}=5.5{}^{\circ}$ by measurement.

**Figure 4 sensors-20-00921-f004:**
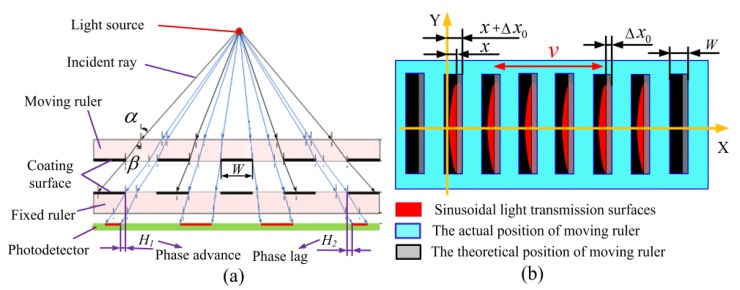
The light transmission model of light source scattering. (**a**) The model of central symmetric phase deviation. (**b**) The average phase shift of light transmission surfaces.

**Figure 5 sensors-20-00921-f005:**
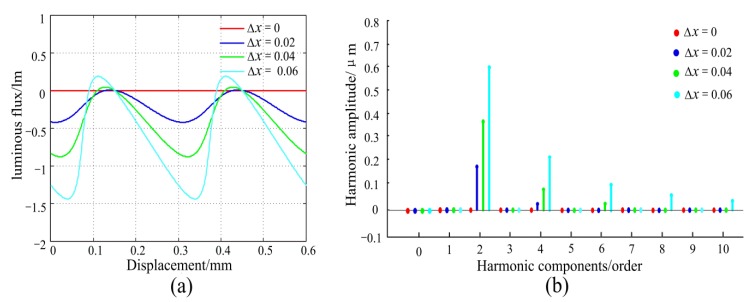
The variation law caused by the phase shift of Δ*x*. (**a**) Error curve. (**b**) Error components.

**Figure 6 sensors-20-00921-f006:**
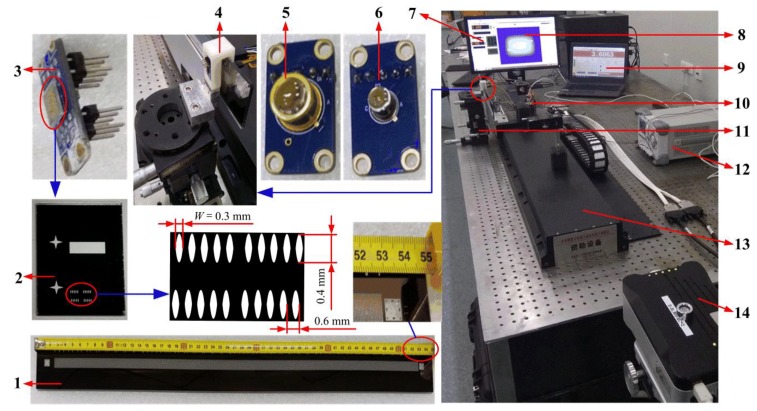
The experimental platform. **1**. Moving ruler; **2**. Fixed ruler; **3**. Photodetector; **4**. The readhead; **5**. LED of $\theta_{1/2}=1.8{}^{\circ}$; **6**. LED of $\theta_{1/2}=5.5{}^{\circ}$; **7**. The interface of the measuring errors for the light-field time grating; **8**. Control system interface of the linear guideway; **9**. Control system interface of the Renishaw XL-80 laser interferometer; **10**. The signal excitation and process system; **11**. The six-axis adjustment mechanism; **12**. Power supply; **13**. The linear guideway; **14**. The Renishaw XL-80 laser interferometer.

**Figure 7 sensors-20-00921-f007:**
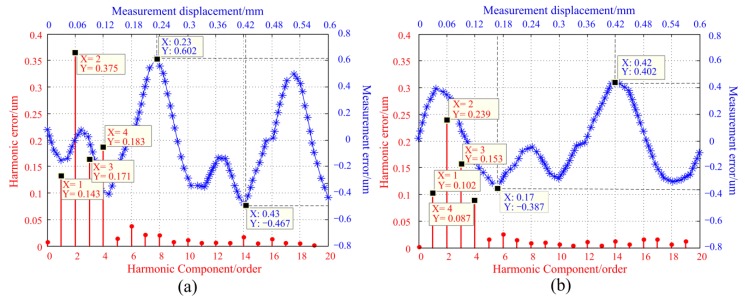
The measurement error curves and the harmonic error components of different light scattering angles. (**a**) $\theta_{1/2}=5.5{}^{\circ}$. (**b**) $\theta_{1/2}=1.8{}^{\circ}$.

**Figure 8 sensors-20-00921-f008:**
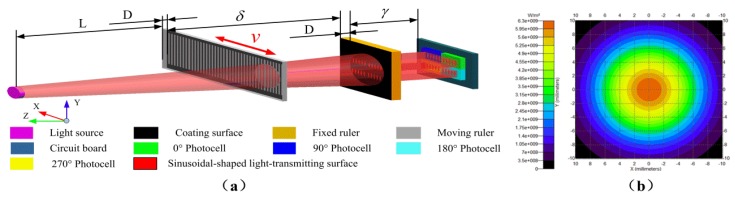
Simulation model and light source simulation. (**a**) The simulation model of the light-field time grating; (**b**) The simulation of 5.5° light source.

**Figure 9 sensors-20-00921-f009:**
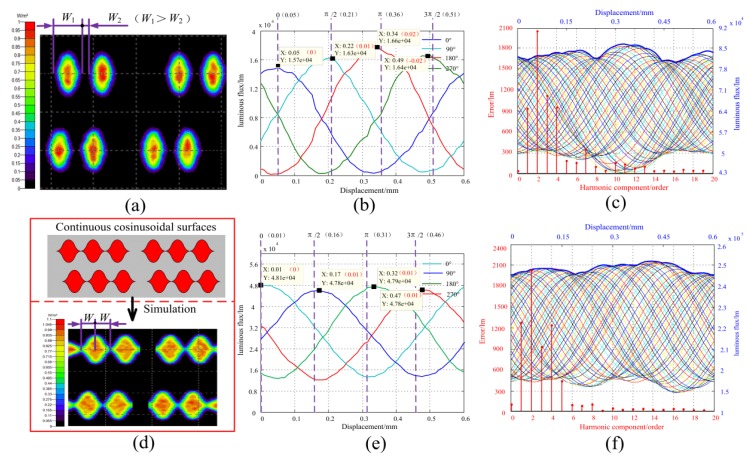
The error analysis of 5.5° light source by simulation. (**a**) The light intensity distribution of the sinusoidal light transmission areas; (**b**) the luminous flux of the sinusoidal light transmission areas; (**c**) the traveling wave signal, error curve and the error components of the sinusoidal light transmission areas; (**d**) the light intensity distribution of the cosinoidal light transmission areas; (**e**) the luminous flux of the cosinoidal light transmission areas; (**f**) the traveling wave signal, error curve and the error components of the cosinoidal light transmission areas.

**Figure 10 sensors-20-00921-f010:**
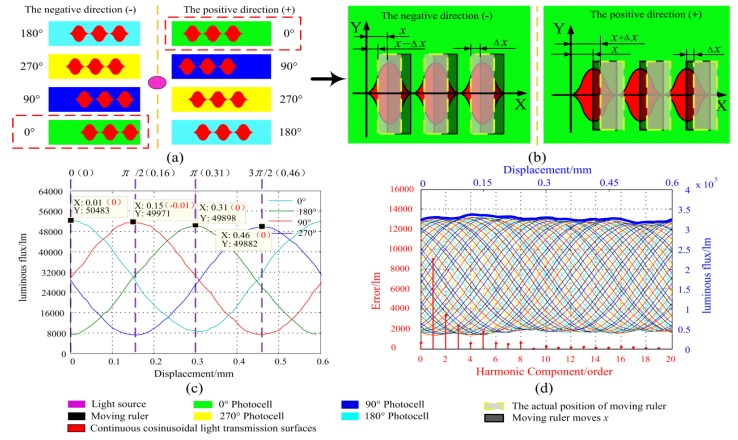
The symmetrical structure model. (**a**) The symmetrical structure and spatial relationship; (**b**) the area changes the law of symmetrical continuous cosinusoidal light transmission surfaces; (**c**) the luminous flux of the symmetrical continuous cosinusoidal light transmission areas; (**d**) the traveling wave signal, error curve and the error components of the symmetrical continuous cosinusoidal light transmission areas.

**Figure 11 sensors-20-00921-f011:**
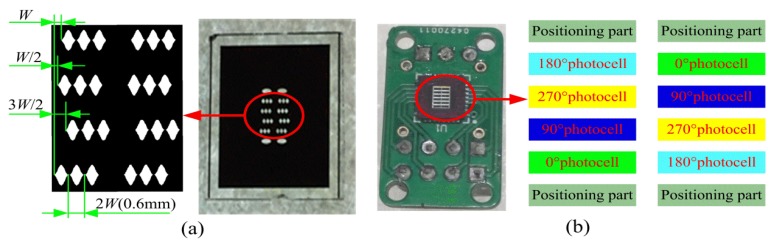
The optimized fixed ruler and photodetector. (**a**) Symmetrical continuous cosinusoidal light transmission surfaces; (**b**) 12-channel active photosensor array.

**Figure 12 sensors-20-00921-f012:**
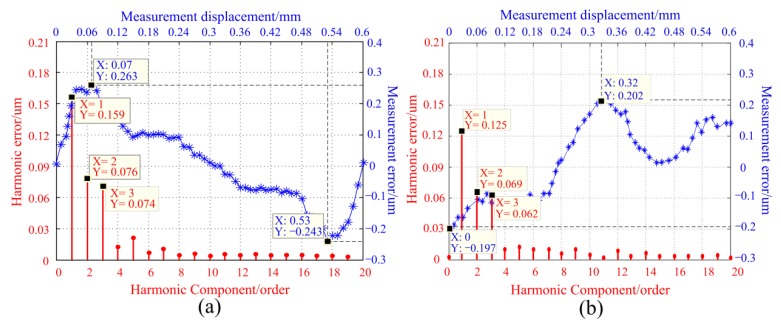
The measurement error curves and the harmonic error components of symmetrical continuous cosinusoidal light transmission surfaces. (**a**) $\theta_{1/2}=5.5{}^{\circ}$; (**b**) $\theta_{1/2}=1.8{}^{\circ}$.

**Figure 13 sensors-20-00921-f013:**
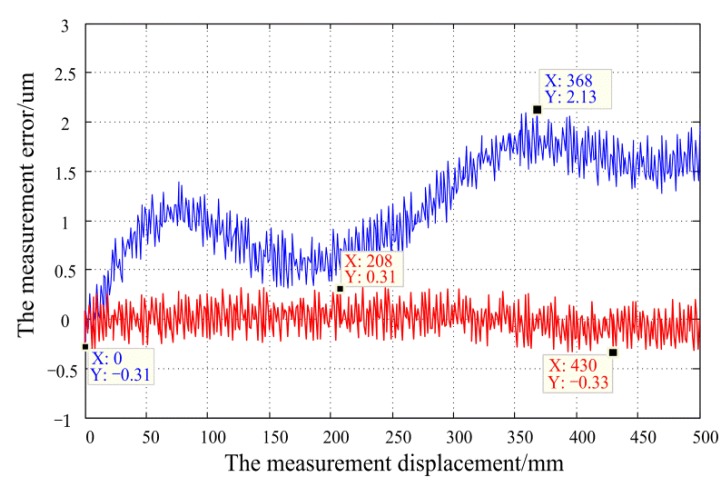
The measurement precision of optimizing structure in a long period. Among, the red line represents for corrected error, and the blue line represents for uncorrected error.

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
