# Peer review of "A High Precision Time Grating Displacement Sensor Based on Temporal and Spatial Modulation of Light-Field"

_sensors, 2020, doi:10.3390/s20030921_

Round 1

Reviewer 1 Report

The manuscript is in good shape except for some minor errors. Please see my suggestions below.

In figure 3a, the fonts in the color bar are too small. It is not clear if Figures 3b and 3c are simulation results or measurements. Authors should explicitly clarify this

Grammatical corrections are needed. For instance, in p6, line 188, it would be "...the variations in the width of the area...."

in p6, line 192, it would be "...phase-lagged."

in p6, line 212, it would be "...spatial phase change...".

and there are other such errors. Please correct them

Author Response

Dear Reviewer

  I am very happy to receive your comments and suggestions. I have made corresponding modifications to your comments, and please see the PDF file.

Thank you!

kind regards,

Author

Reviewer 2 Report

In this study, the authors propose a new approach to measure displacement  through time grating based on the modulation of light-field . In particular the authors demonstrate by theoretical models and experimental tests that the error given by light uniformity and scattering angle can be greatly reduced, achieving a sub-micrometric accuracy in a large measurement range.  The proposed model  allows to achieve high accuracy and at the same time it is not limited by the manufacturing costs to sustain to fabricate  gratings for high-precision measurements.

The relevance of the sensing technology appears to be significant from a technological perspective. The manuscript is well structured and clearly written. The assessment of the sensing performance  in light of the error analysis and reduction is rigorously performed and the conclusions are adequately supported both by  the theoretical simulations and the experimental findings.  I believe the manuscript could be further improved by addressing the following concerns that would render it more interesting and impactful to a broader  scientific audience in the field of sensors:

The novelty of the sensing method should be further elucidated. The authors mention the accuracy improvement in the conclusions. However, I would recommend to expand the comparison with respect to conventional grating sensors in the discussion in order to better highlight the relevance of the proposed system. While the authors provide a validation of their time grating based sensor to measure displacement and its operation principle, the impact of the developed technology should be clarified in the context of potential industrial/technological applications in which such sensing system would outperform existing technology.

Author Response

Dear Reviewer

  I am very happy to receive your comments and suggestions. I have made corresponding modifications to your comments, and please see the PDF file.

Thank you !

kind regards,

Author

Reviewer 3 Report

This paper titled “A high precision time grating displacement sensor 2 based on temporal and spatial modulation of light-field,” presented an interesting work.  This technology potentially could overcome the bottleneck on the scale manufacture of laser encoder.  The current version of this article is very close to be published.  Here are some comments.

At line 204, in Eqn (7), the expression in cosine function looks not in radian dimension. Please check. At line 211, in Eqn (8), there is the same problem like comment 1. At lines 165 and 166, some symbols are in wrong format. At line 243, some symbols are in wrong format. At line 265, some symbols are in wrong format. With Figure 15, why can you say the accuracy is 0.64 um in the range of 500 mm? Why is the error band 0.4 um?  Can you say more about the 0.4 um error band?  And what does the red line mean?

Author Response

(The authors gave the same response as above.)
